# Medium Bandgap Polymers for Efficient Non-Fullerene Polymer Solar Cells—An In-Depth Study of Structural Diversity of Polymer Structure

**DOI:** 10.3390/ijms24010522

**Published:** 2022-12-28

**Authors:** Shimiao Zhang, Dong Hwan Son, Rahmatia Fitri Binti Nasrun, Sabrina Aufar Salma, Hongsuk Suh, Joo Hyun Kim

**Affiliations:** 1Department of Chemistry and Chemistry Institute for Functional Materials, Pusan National University (PNU), Busan 46241, Republic of Korea; 2CECS Research Institute, Core Research Institute, Busan 48513, Republic of Korea; 3Department of Polymer Engineering, Pukyong National University, Busan 48513, Republic of Korea

**Keywords:** non-fullerene acceptor, indandione, polymer solar cell, thiophene bridges, inverted solar cell

## Abstract

A series of medium bandgap polymer donors, named poly(1-(5-(4,8-bis(5-(2-ethylhexyl)-4-fluorothiophen-2-yl)benzo [1,2-*b*:4,5-*b*′]dithiophen-2-yl)thiophen-2-yl)-5-((4,5-dihexylthiophen-2-yl)methylene)-3-(thiophen-2-yl)-4*H*-cyclopenta[c]thiophene-4,6(5*H*)-dione) (IND-T-BDTF), poly(1-(5-(4,8-bis(5-(2-ethylhexyl)-4-fluorothiophen-2-yl)benzo [1,2-*b*:4,5-*b*′]dithiophen-2-yl)-4-hexylthiophen-2-yl)-5-((4,5-dihexylthiophen-2-yl)methylene)-3-(4-hexylthiophen-2-yl)-4*H*-cyclopenta[*c*]thiophene-4,6(5*H*)-dione (IND-HT-BDTF), and poly(1-(5-(4,8-bis(5-(2-ethylhexyl)-4-fluorothiophen-2-yl)benzo [1,2-*b*:4,5-*b*′]dithiophen-2-yl)-6-octylthieno [3,2-*b*]thiophen-2-yl)-5-((4,5-dihexylthiophen-2-yl)methylene)-3-(6-octylthieno [3,2-*b*]thiophen-2-yl)-4*H*-cyclopenta[*c*]thiophene-4,6(5*H*)-dione (IND-OTT-BDTF), are developed for non-fullerene acceptors (NFAs) polymer solar cells (PSCs). Three polymers consist of donor-acceptor building block, where the electron-donating fluorinated benzodithiophene (BDTF) unit is linked to the electron-accepting 4*H*-cyclopenta[*c*]thiophene-4,6(5*H*)-dione (IND) derivative via thiophene (T) or thieno [3,2-*b*]thiopene (TT) bridges. The absorption range of the polymer donors based on IND in this study shows 400~800 nm, which complimenting the absorption of Y6BO (600~1000 nm). The PSC’s performances are also significantly impacted by the π-bridges. NFAs inverted type PSCs based on polymer donors and Y6BO acceptor are fabricated. The power conversion efficiency (PCE) of the device based on IND-OTT-BDTF reaches up to 11.69% among all polymers with a short circuit current of 26.37 mA/cm^2^, an open circuit voltage of 0.79 V, and a fill factor of 56.2%, respectively. This study provides fundamental information on the invention of new polymer donors for NFA-based PSCs.

## 1. Introduction

For the last several decades, tremendous efforts have been made to develop highly efficient PSCs to solve energy-related issues such as global warming caused by the consumption of fossil oils. PSCs have been one of the prospective fields due to their lightweight, low cost, flexibility, and scalability [1,2,3,4]. PSCs are often created by combining an n-type electron acceptor (A) and p-type polymer donor (D) to create a bulk heterojunction (BHJ), which can create a significant interfacial area between D and A and improve charge generation [5,6,7]. Recently, polymer donors based on PM6 [8] or polymers based on quinoxaline derivatives [9] with high PCE have been continuously reported. The PCEs of PSCs based on those polymers with non-fullerene acceptors (NFAs), such as Y6BO, might reach up to 19% [8,9,10,11,12,13,14,15,16]. The performances of NFA-based PSCs were generally much better than those of fullerene-based PSCs due to the benefits of NFAs, including their enhanced light absorption, tunable optical, and electrochemical properties, and adequate morphological control [2,8]. In addition, the absorption of low bandgap NFAs is a very beneficial complementary to the medium bandgap polymer donors and reached PCE up to 15%. Numerous polymer donors with a backbone architecture consisting of D and A electron units have been created for PSCs. Due to their planar structure and good charge transporting qualities, benzodithiophene (BDT) and fluorine-substituted benzodithiophene (BDTF) units have been widely used as the good electron-donating group among many high-performance polymer donors [9,17,18]. Quinoxaline (Qx), benzo [1,2-*c*:4,5-*c*0]dithiophene-4-8-dione (BDD), benzotriazole (BTA), and other electron-deficient components have also been competitively utilized to build effective D-A type polymer donors for NFA-based PSCs [19,20,21,22,23,24,25,26,27,28,29,30].

Recently, 2-((*Z*)-2-((10-((*Z*)-((*E*)-1-(cyano(isocyano)methylene)-5,6-difluoro-3-oxo-1*H*-inden-2(3*H*)-ylidene)methyl)-12,13-bis(2-butyloctyl)-3,9-diundecyl-12,13-dihydro-[1,2,5]thiadiazolo [3,4-*e*]thieno [2″,3″:4′,5′]thieno [2′,3′:4,5]pyrrolo [3,2-*g*]thieno [2′,3′:4,5]thieno [3,2-*b*]indol-2-yl)methylene)-5,6-difluoro-3-oxo-2,3-dihydro-1*H*-inden-1-ylidene)malononitrile (Y6BO) and 2,2′-[[6,6,12,12-Tetrakis(4-hexylphenyl)-6,12-dihydrodithieno [2,3-*d*:2′,3′-*d*′]-s-indaceno [1,2-*b*:5,6-*b*′]dithiophene-2,8-diyl]bis[methylidyne(3-oxo-1*H*-indene-2,1(3*H*)-diylidene)]]bis[propanedinitrile] (ITIC) derivatives are widely used for acceptors for NFA-based PSCs. Both NFAs have 2-(3-oxo-2,3-dihydro-1*H*-inden-1-ylidene)malononitrile end-group derivatives. Thus, 2-(3-oxo-2,3-dihydro-1*H*-inden-1-ylidene)malononitrile (indandione-CN) (Figure 1) derivatives without malononitrile group (1*H*-indene-1,3(2*H*)-dione) (indandione) will be a very good electron-acceptor unit for medium-bandgap conjugated polymer donors.

Inspired by the structure of ITIC and Y6BO derivatives, we synthesized a series of medium bandgap 4*H*-cyclopenta[*c*]thiophene-4,6(5*H*)-dione (IND) conjugated polymers depicted in Figure 1, in which the BDTF is linked to the IND via thiophene (T), hexylthiophene, or thieno [3,2-*b*]thiopene (TT) as the π-bridge. Depending on the conjugated bridges, the polymer main chain will generate varied interactions between the donor and acceptor units, which will provide a different electronic structure for the copolymers. D-π-A conjugated copolymers with various conformational, electrochemical, charge transport, optical, and photovoltaic characteristics will therefore be produced. In this study, IND-based polymer donors displayed an absorption range of 400~800 nm well complemented with the absorption of Y6BO (600~1000 nm). In addition, the polymer donors may exhibit very good compatibility with NFA in the blend due to the structural similarity. Inverted PSCs type with a configuration of ITO/ZnO/polymer donor:Y6BO/MoO_3_/Ag were fabricated and tested. The PSC of the IND-OTT-BDTF device exhibited the highest PCE of 11.7% with a short circuit current ($J_{sc}$) of 26.4 $\mathrm{mA}/{\mathrm{cm}}^{2}$, an open circuit voltage ($V_{oc}$) of 0.79 V, and a fill factor (FF) of 56.2%, respectively.

## 2. Results and Discussion

### 2.1. Synthesis and Characterization

The synthesis procedure thorough description is provided in Figure 1 which illustrates the synthesis of monomers and polymers. The structure of each compound was confirmed by ^1^H and ^13^C NMR analysis (Appendix A). Compound **5** was prepared by the Knoevenagel condensation reaction between IND and **4**. Compounds **6**, **7**, and **8** were prepared by the Stille coupling reaction between compound **5** and corresponding tributyl tin compounds with fairly high yields of 81.5, 82.9, and 59.4%, respectively. The Stille coupling reactions between BDTF and M1, M2, or M3 afforded the polymers IND-T-BDTF, IND-HT-BDTF, and IND-OTT-BDTF, respectively. In chlorinated organic solvents such as chloroform and chlorobenzene, the polymers were completely soluble. The number average molecular (M_n_) weight/polydispersity index (PDI) of polymers IND-T-BDTF, IND-HT-BDTF, and IND-OTT-BDTF were 4.3 kDa/2.15, 7.5 kDa/1.62, and 22.5 kDa/2.42, respectively. Due to many insoluble parts after Soxhlet extraction by chloroform, IND-T-BDTF and IND-HT-BDTF possessed low molecular weight. According to Appendix A, the polymers had good thermal stability, IND-T-BDTF, IND-HT-BDTF, and IND-OTT-BDTF start to decompose at temperatures (T_d_, 5% weight loss) of 362, 361, and 385 °C, respectively. In differential scanning calorimetry, no observable melting behavior or glass transition was seen.

### 2.2. Optical and Electrochemical Behaviors

The absorption spectra of three polymer films are depicted in Figure 2 and their optical characteristics are summarized in Table 1. IND-T-BDTF film showed two broad absorption bands. The backbone’s π–π* transition corresponds to a former absorption band at 400~500 nm. An absorption band in the longer wavelength region refers to intramolecular charge transfer (ICT) between the donor (BDTF) and acceptor (IND), a typical property of polymers with D–A arrangement, at a longer wavelength region (550~800 nm). The maximum absorption wavelengths of π–π* transition of polymer films were almost identical. The maximum absorption wavelength of ICT in IND-HT-BDTF film exhibited at 636 nm, which is shorter than that of IND-T-BDTF. This is due to the dihedral angle (see Appendix A) between the 3-hexylthiophene ring and IND of IND-HT-BDTF being larger than that of the IND-T-BDTF. The maximum absorption wavelength ICT in IND-OTT-BDTF film appeared at 668 nm, which is red-shifted than those of the IND-HT-BDTF and IND-T-BDTF. This suggests that thieno [3,2-*b*]thiopene (TT) exhibited higher aromaticity compared to thiophene, indicating increased electron delocalization in the polymer backbone [31,32]. The optical/electrochemical bandgaps (Table 1) of IND-T-BDTF, IND-HT-BDTF, and IND-OTT-BDTF figured out from the absorption edge were 1.65/1.76, 1.65/1.78, and 1.65/1.58 eV, respectively, which are almost identical. The absorption coefficients of IND-T-BDTF, IND-HT-BDTF, and IND-OTT-BDTF films at the ICT maximum wavelengths were 3.29 × 10^4^, 3.05 × 10^5^, and 4.30 × 10^5^ cm^−1^, respectively. Due to the strong ICT behavior in the polymer with the electron-rich and planar TT group, it is observed that the IND-OTT-BDTF film had the maximum absorption coefficient when compared to those of IND-T-BDTF and IND-HT-BDTF. According to Figure 2a, which includes energy absorption from the near-IR region to UV-Vis, the absorption range of polymers is complementary to those of Y6BO acceptors.

The energy levels of the polymers were measured by cyclic voltammetry. According to Appendix A, the HOMO/LUMO energy levels of IND-T-BDTF, IND-HT-BDTF, and IND-OTT-BDTF were −5.35/−3.59, −5.36/−3.58, and −5.19/−3.61 eV, respectively. Despite the increased aromaticity of TT in IND-OTT-BDTF compared with thiophene, the energy level of IND-OTT-BDTF is almost identical to those of IND-T-BDTF and IND-HT-BDTF.

The polymers and the materials used in this research with those energy level diagrams are displayed in Figure 2b. Facile charge separation and transport processes are expected to happen in the devices. We also performed photoluminescence (PL) experiments to further investigate the exciton dissociation and charge transfer behavior from polymer donors to Y6BO. The PL spectra in Appendix A showed broad emission at 700~850 nm. The fact that the PL emissions from polymer blend films containing Y6BO were almost quenched shows that the exciton dissociation and charge transfer in the blend films have successfully taken place.

The frontier molecular orbitals of the polymers were determined from Density functional theory (DFT) at the B3LYP/6-31G** level of the Gaussian 09 (ver.9.5) software (Wallingford, CT, UK) [33]. For making computation easier, the polymer alkyl chains were represented by methyl groups, and two repeating units to represent the polymer itself. Wave functions in the HOMO state of the polymers were delocalized along the backbone and BDTF unit, as shown in Appendix A. However, the LUMO wave functions were localized in the IND unit. IND-T-BDTF, IND-HT-BDTF, and IND-OTT-BDTF have HOMO/LUMO energy levels of −4.91/−2.81, −4.89/−2.74, and −4.91/−2.78 eV, respectively. The incorporation of an extended π-bridge did not affect the energy levels of the polymers.

### 2.3. Photovoltaic Property

Inverted-type PSCs with a configuration of ITO/ZnO/donor polymers:Y6BO/MoO_3_/Ag were evaluated to know the photovoltaic performances of polymers. First, several processing parameters, including the active layer thickness and D–A blend ratios were investigated for observing how they affect the photovoltaic performance. The optimum blend ratio between the polymers and Y6BO was performed at 3:4 and with 130 nm for IND-T-BDTF and 120 nm for IND-HT-BDTF and IND-OTT-BDTF active layer optimum thickness achieved (Appendix A). Figure 3a,b display the current density (J) vs applied voltage (V) curves under illumination and dark with photovoltaic parameters of the devices, and their optimum processing conditions are summarized in Table 2. 

Among devices based on polymer donors, the device based on IND-OTT-BDTF had the greatest PCE, at 11.7%. The devices based on IND-T-BDTF, IND-HT-BDTF, and IND-OTT-BDTF have corresponding $V_{OC}$ values of 0.80, 0.87, and 0.79 V. The $V_{OC}$ trend in all polymers agrees well with their HOMO. When compared to the device based on IND-T-BDTF (11.9 mA/cm^2^), the $J_{sc}$ of IND-HT-BDTF and IND-OTT-BDTF were respectively enhanced to 21.9 and 26.4 mA/cm^2^. This conforms with the absorption coefficient of polymers. Incident photon-to-electron conversion efficiency (IPCE) curves, which range from 300 to 900 nm, are shown in Appendix A. The polymers prove to complement the solar light by the Y6BO absorption. The calculated $J_{sc}$ values of the devices derived from the IPCE spectra were highly correlated to the $J_{sc}$ data under 1.0 sun cconditions. The fill factor (FF) was slightly improved from 33.5% to 37.9% by the addition of extra hexyl groups on thiophene linkage in IND-HT-BDTF. This is apparently because of the better solubility of IND-HT-BDTF than that of IND-T-BDTF. It was further improved to 56.2% in IND-OTT-BDTF due to the TT bridge exhibiting the more extended π-conjugation. Simultaneously improvement in the $V_{OC}$, $J_{sc}$, and FF in IND-OTT-BDTF can result in the significant enhancement of the PCE. Consequently, IND-OTT-BDTF-based device showed the best PCE. Under the dark condition of J-V curves (inset of Figure 3b), the series resistance ($R_{s}$) and shunt resistance ($R_{sh}$) of the devices were calculated. The $R_{s}$ of IND-OTT-BDTF were 2.21 Ω cm^2^ was the smallest compared to the devices with IND-T-BDTF (9.64 Ω cm^2^) and IND-HT-BDTF (7.82 Ω cm^2^). Additionally, among the devices based on polymers, the $R_{sh}$ data of the IND-OTT-BDTF-based device (0.38 kΩ cm^2^) was the highest. The values of $J_{sc}$ and FF for related devices showed a good match with the trend of $R_{s}$ and $R_{sh}$.

To further determine the charge carrier transport capabilities, electron- and hole-only devices with structures of ITO/ZnO (25 nm)/donor:Y6BO/LiF/Al (100 nm) and ITO/PEDOT:PSS (35 nm)/donor:Y6BO/Au (50 nm) were created and evaluated. The J-V curves of matching devices (see Appendix A) fit the law and displayed the characteristics of space charge limited current (SCLC) behavior refers to the Mott-Gurney law [34,35]. IND-T-BDTF, IND-HT-BDTF, and IND-OTT-BDTF have the hole mobility of 1.55 × 10^−4^, 3.16 × 10^−4^, and 5.55 × 10^−4^ cm^2^ V^−1^ s^−1^, respectively, while their electron mobilities were 1.71 × 10^−4^, 3.23 × 10^−4^, and 5.01 × 10^−4^ cm^2^ V^−1^ s^−1^, respectively. This result indicates better charge transport and extraction in the devices based on IND-OTT-BDTF, which agrees with the higher J_sc_ and FF values of the corresponding devices. 

Furthermore, the carrier transporting and collecting behavior of the devices, photocurrent density ($J_{ph}$) as a function of effective voltage ($V_{eff}$) relationship was calculated (Figure 4). Here, the $J_{ph}$ is defined as $J_{D}-J_{L}$, $J_{D}$ and $J_{L}$ are the current density under dark and illumination conditions, respectively. Charge carrier transporting and collecting probability ($J_{ph}/J_{sat}$) data of the devices based on IND-T-BDTF, IND-HT-BDTF, and IND-OTT-BDTF were estimated to be 50.9, 64.9, and 88.9%, respectively, proving that IND-OTT-BDTF based device showed the best charge carrier transporting and collecting behavior. The maximum exciton generation rate ($G_{max}$) is correlated to the absorption ability of the active layer. The $G_{max}$ values (defined as $J_{ph}/q\cdot L$, q and L are electron charge and active layer thickness) of devices based on IND-T-BDTF, IND-HT-BDTF, and IND-OTT-BDTF were 4.96 × 10^26^, 1.49 × 10^27^, and 1.50 × 10^27^ cm^3^ s^−1^, respectively, which are in line with the enhancement of the polymer films absorption coefficient [36].

We measured the devices $J_{sc}$ and $V_{OC}$ vs. light intensity to better understand the charge recombination mechanisms. From the relationship between the $J_{sc}$ vs. illuminated light intensity ($P_{light}$) (expressed by $J_{sc}$ = ($P_{light}$)^α^), the bimolecular recombination process can be observed. According to Figure 5a, the α values of IND-T-BDTF, IND-HT-BDTF, and IND-OTT-BDTF based devices were 0.76, 0.89, and 1.05, respectively. As for the device with IND-OTT-BDTF, the bimolecular recombination process was minimized in the device [37]. Thus, IND-T-BDTF shows poorer PCE than those based on IND-HT-BDTF, and IND-OTT-BDTF devices. Additionally, using the formula $V_{OC}$ = (nkT/q) × ln ($P_{light}$) (where k, q, and T are the Boltzmann constant, elementary charge, and the temperature in Kelvin), it is possible to determine the device trap-assisted recombination. If *n* becomes 1, the band-to-band recombination process is dominated in the device. Trap-assisted recombination mechanism predominates in the devices when *n* is close to 2 [21]. In contrast to the device with IND-T-BDTF (*n* = 1.47) which demonstrated the most undesired trap-assisted recombination process, IND-HT-BDTF and IND-OTT-BDTF had *n* values of 1.02 and 1.18, respectively.

### 2.4. Morphology Study

The molecular ordering information is very important for understanding the overall photovoltaic properties of PSCs. To understand the ordering features of the active layers, we measured grazing incidence wide-angle X-ray scattering (GIWAXS). Figure 6 showed GIWAXS images (Figure 6a,c) and direction line cuts in-plane (IP) and out-of-plane (OOP) (Figure 6b,d) of neat polymers and polymer:Y6BO blending film. GIWAXS film was prepared in the same way as the preparation of devices on the silicon wafer. As shown in Figure 6b (OOP direction), IND-T-BDTF and IND-OTT-BDTF films exhibited a broad (010) peak at 1.71 and 1.65 Å^−1^, respectively, corresponding to π–π (intermolecular) stacking distances of 3.67 and 3.81 Å, respectively. As for IND-HT-BDTF film, very weak (010) peak at 1.68 Å^−1^ (3.74 Å). The intermolecular stacking distances were increased in the order of IND-T-BDTF < IND-HT-BDTF < IND-OTT-BDTF due to the alkyl substituents on the π-bridges on T and TT. By a peak (010) in the OOP direction cut, it is preferable to have the face-on orientation to the surface. This means that the vertical charge transport is favorable in the device [38]. The lamellar domain is represented by a broad (100) peak in the IND-T-BDTF, IND-HT-BDTF, and IND-OTT-BDTF at 0.274 (22.9), 0.266 (23.6), and 0.267 (23.5) Å^−1^ (Å) through the IP direction, respectively. Alkyl substituents on the π-bridges such as thiophene (T) and TT also affect the lamellar domain spacing distances. As shown in Figure 6d, a broad peak at 1.70 (3.70) Å^−1^ (Å) in OOP directions appeared, which is almost the same as the peak in Y6BO film (1.75 Å^−1^) (See Appendix A). Interestingly, two unknown scattering patterns in IP directions at 1.36 and 1.70 Å^−1^ for the blend films were found. The strength of the (010) peak along the OOP direction in blend films is more pronounced than the peak in neat polymer films. This indicates that the Y6BO acceptor is the key factor for the face-on orientation in the blend films. Although blend films primarily refer to the face-on molecular packing orientation of the Y6BO acceptor, the blend films may also form in a favored face-on orientation owing to D–A strong intermolecular interactions. Moreover, to know the active layer morphology, we also used transmission electron microscopy (TEM) and atomic force microscopy (AFM). The TEM images of the active layer (Appendix A) showed that the IND-HT-BDTF and IND-OTT-BDTF blend films may create bicontinuous interpenetrating networks and superior nanoscale phase separation than the IND-T-BDTF blend films, respectively. Therefore, by effective charge-separation or transport, phase separation is preferred in the active layer based on IND-HT-BDTF and IND-OTT-BDTF can promote higher PCE of the related PSCs. The AFM height and phase images (Appendix A) of the blend films based on IND-T-BDTF, IND-HT-BDTF, and IND-OTT-BDT exhibit the root-mean-square (RMS) surface roughness of 2.42, 2.02, and 1.18 nm, respectively. The AFM morphologies of the IND-OTT-BDTF-based blend film are beneficial for exciton diffusion and dissociation to achieve higher Jsc and FF.

## 3. Materials and Methods

### 3.1. Materials and Instruments

2,2′-((2*Z*,2′*Z*)-((12,13-bis(2-butyloctyl)-12,13-dihydro-[1,2,5]thiadiazolo [3,4-*e*]thieno [2″,3″:4′,5′]thieno [2′,3′:4,5]pyrrolo [3,2-*g*]thieno [2′,3′:4,5]thieno [3,2-*b*]indole-2,10-diyl)bis (methanylylidene))bis(5,6-difluoro-3-oxo-2,3-dihydro-1*H*-indene-2,1-diylidene)) dimalononitrile (Y6BO) [9], and (4,8-bis(5-(2-ethylhexyl)-4-fluorothiophen-2-yl)benzo-[1,2-*b*:4,5-*b*′]dithiophene-2,6-diyl)bis(trimethylstannane)(BDTF) [23] were synthesized according to previous reports. 4,6-Dibromo-1*H*,3*H*-thieno [3,4-*c*] furan-1,2-dione was purchased form Sunatech. All other chemicals used in this work were purchased from Sigma Aldrich Co. (St. Louis, MO, USA) and Alfa Aesar (A Johnson Matthey Company, Haverhill, MA, USA), and used without any further purification unless otherwise described. The ^1^H and ^13^C NMR spectra were measured with a JEOL JNM ECP-400 spectrometer. UV visible spectra were recorded on a JASCO V730 UV/Vis spectrophotometer. Matrix-assisted laser desorption/ionization time-of-flight (MALDI-TOF) spectroscopy was conducted by using a Bruker Ultraflex spectrometer. Gel permeation chromatography (GPC) was measured on an Agilent 1200 series instrument with THF as the eluent. The thermogravimetric analysis (TGA) was carried out under the N_2_ atmosphere at a heating rate of 10 °C/min with TA Instrument Q600 (PH407 PUSAN KBSI). Cyclic voltammetry (CV) measurements were carried out by using a VersaSTAT3 potentiostat (Princeton Applied Research) with tetrabutylammonium hexafluorophosphate (0.1 M, Bu_4_NPF_6_) as the electrolyte in acetonitrile. The films thickness was measured with an Alpha-Step IQ surface profiler (KLA-Tencor Co., Milpitas, CA, USA) Grazing-incidence wide-angle X-ray scattering (GIWAXS) spectra were obtained on the 3C beamline with 13 keV (λ = 0.123 nm) X-ray irradiation source and the beam size of 300 μm (height) × 23 μm (width) in the Pohang Accelerator Laboratory (PAL). A two-dimensional charge-coupled device detector (Mar165 CCD) was used, and the distance from the sample to the detector was 0.2 m. The X-ray beam angle of the incidence was chosen such that the beam would penetrate the entire active layer while minimizing scattering from the substrate: ~0.12°. The samples were partially completed devices so that the entire exposed surface is composed of an active layer on the Si wafer and were examined under ambient. Preparation of film for GIWAXS was followed the same as the preparation of the active layer. The ZnO layer was deposited on the Si wafer by sol-gel process giving a film of 25-nm-thick. The polymer or blended polymer film (polymeric donor and Y6BO acceptor) was fabricated by spin-coating in chloroform with 0.5% of 1-chloronaphthalene (CN) as a processing additive. Then the film was annealed at 100 °C for 10 min in the glove box. The scattering vector (q) and d spacing (d) were calculated from the equation: q = 4π sin (ϕ)/λ and q = 2π/d. Photoluminescence spectra of the polymers and blended films were obtained by a HOMOBA (fluolog-QM). The blend morphology was examined by the transmission electron microscope (HITACHI Hightech. HT-7800).

### 3.2. Synthesis of Monomers and Polymers

^1^H and ^13^C NMR spectra of synthesized compounds are displayed in Appendix A).

#### 3.2.1. Synthesis of 1,3-Dibromo-4*H*-cyclopenta[*c*]thiophene-4,6(5*H*)-dione (IND)

A portion of triethylamine (1 mL) and 0.240 g (1.8 mmol) ethylaceto acetate were added to the solution of 4,6-dibromo-1*H*,3*H*-thieno [3,4-*c*] furan-1,2-dione (0.406 g, 1.3 mmol) in 1 mL of acetic anhydride under nitrogen. Then the reaction mixture was refluxed at 65 °C overnight. After cooling down to room temperature the mixture was poured into diluted HCl under ice-bath conditions and extracted with MC. The organic phase was evaporated and the mixture of the residue in concentrated HCl was refluxed at 60 °C for 2 h. The mixture was extracted with MC, dried over MgSO_4_ and the solvent was evaporated by reduced pressure. The obtained pink solid was purified by silica gel column chromatography using MC/hexane (10:1) as eluent, pink solid was obtained (0.207 g, 51.0%). ^1^H NMR (400 MHz, CDCl_3_, ppm): δ 3.51 (s, 2H). ^13^C NMR (100 MHz, CDCl_3_, ppm): δ 187.07, 145.52, 113.05, 53.29. LRMS (*m*/*z*, EI+) calcd for C_7_H_2_Br_2_O_2_S 309.966 found 309.057.

#### 3.2.2. Synthesis of 2-Bromo-3-Hexylthiophene (**2**)

After 3 g (17.8 mmol) of 3-hexylthiophene was dissolved in 40 mL of tetrahydrofuran (THF), 3.49 g (19.6 mmol) of *N*-bromosuccinimide (NBS) was slowly added under ice-bath condition. The reaction was kept for 3 h at room temperature and monitored by TLC. The reaction was ended by adding 100 mL of water and then the mixture was extracted with 100 mL of diethyl ether. The organic phase was collected and washed several times with brine. After drying over MgSO_4_, the solvent was evaporated under reduced pressure. Finally, the product was purified by column chromatography using hexane as eluent, transparent oil was obtained (4.10 g, 93.4%). ^1^H NMR (400 MHz, CDCl_3_, ppm): δ 7.20 (d, 1H), 6.82 (d, 1H), 2.60 (t, 2H), 1.61 (m, 2H), 1.35 (m, 6H), 0.93 (t, 3H). ^13^C NMR (100 MHz, CDCl_3_, ppm): δ 142.06, 128.34, 125.23, 108.93, 31.78, 29.86, 29.54, 29.05, 22.76, 14.25. LRMS (*m*/*z*, EI+) calcd for C_10_H_15_BrS 247,191 found 247.102.

#### 3.2.3. Synthesis of 2,3-Dihexylthiophene (**3**)

In a two-necked flask, 5.41 g (21.9 mmol) of 2-bromo-3-hexylthiophene (**2**) and 0.59 g (1 mmol) of Ni(dppp)Cl_2_ were dissolved in 25 mL of distilled THF. Hexyl-MgBr was slowly added to the reaction mixture at the ice bath temperature. Then the mixture was refluxed under nitrogen conditions overnight. A saturated solution of ammonium chloride was added to end the reaction and the reaction mixture was further extracted with hexane. After washing several times with brine, the mixture was dried over MgSO_4_ and the solvent was evaporated by a rotary evaporator. The product was purified by column chromatography using hexane as eluent, yellow oil was obtained. (4.90 g, 88.0%). ^1^ H NMR (400 MHz, CDCl_3_, ppm): δ 7.03 (d, 1H), 6.82 (d, 1H), 2.72 (t, 2H), 2.51 (t, 2H), 1.63 (m, 2H), 1.55 (m, 2H), 1.31 (m, 12H), 0.90 (t, 6H). ^13^C NMR (100 MHz, CDCl_3_, ppm): δ 138.89, 137.78, 128.76, 120.96, 32.03, 31.84, 31.73, 30.94, 29.81, 29.30, 29.14, 28.31, 27.87, 22.73, 22.70, 14.19.

#### 3.2.4. Synthesis of 4,5-Dihexylthiophene-2-Carbaldehyde (**4**)

A mixture of DMF (6.95 g, 95.1 mmol) and POCl_3_ (14.58 g, 95.1 mmol) was stirred at 0 °C for 30 min to form the Vilsmeier reagent; the Vilsmeier reagent was slowly added to a solution of 6.15 g (24.4 mmol) of 2,3-dihexylthiophene (**3**) in 48 mL of dichloroethane. The reaction mixture was refluxed overnight under an N_2_ environment. After cooling down to room temperature, an aqueous NaHCO_3_ solution was added. The mixture was extracted with MC, dried over with MgSO_4_ and the solvent was evaporated by reduced pressure. A brown oil was further purified by silica gel column chromatography using MC/hexane (6:4) to afford the product as a yellow oil (6.60 g, 96.0%). ^1^H NMR (400 MHz, CDCl_3_, ppm): δ 9.78 (s,1H), 7.03 (s, 1H), 2.77 (t, 2H), 2.52 (t, 2H), 1.66 (m, 2H), 1.57 (m, 2H), 1.31 (m, 12H), 0.89 (t, 6H). ^13^C NMR (100 MHz, CDCl_3_, ppm): δ 182.69, 151.88, 140.11, 139.47, 138.48, 31.73,31.60, 31.29, 30.52, 29.12, 29.01, 28.80, 28.13, 22.67, 22.62, 14.13. LRMS (*m*/*z*, EI+) calcd for C_17_H_28_OS 280.466, found 280.487.

#### 3.2.5. Synthesis of 1,3-Dibromo-5((4,5-dihexylthiophen-2-yl)methylene)4*H*cyclopenta[*c*]thiophene-4,6(5*H*)-dione (**5**)

A mixture of compound **4** (0.306 g, 1.09 mmol), IND (0.402 g, 1.30 mmol), and 6 drops of pyridine in 15 mL of anhydrous chloroform was refluxed at 65 °C for overnight under nitrogen conditions. Water was poured into the reaction mixture, then extracted with MC and the organic layer was dried over MgSO_4_. After removing the solvent by reduced pressure, the crude product was further purified by column chromatography using MC/hexane (7:3) as eluent, yellow solid was obtained (0.357 g, 57.0%). ^1^H NMR (400 MHz, CDCl_3_, ppm): δ 7.88 (s,1H), 7.79(s, 1H), 2.82 (t, 2H), 2.54 (t, 2H), 1.72 (m, 2H), 1.58 (m, 2H), 1.31 (m, 12H), 0.88 (t, 6H). ^13^C NMR (100 MHz, CDCl_3_, ppm): δ 181.38, 181.02, 159.76, 145.78, 143.68, 143.23, 141.84, 139.93, 133.65, 129.12, 111.80, 111.75, 31.72, 31.60, 31.32, 30.49, 29.42, 29.24, 29.18, 27.89, 22.68, 22.62, 14.17, 14.14. LRMS (*m*/*z*, EI+) calcd for C_24_H_28_Br_2_O_2_S_2_ 570.121, found 570.118.

#### 3.2.6. Synthesis of 5-((4,5-Dihexylthiophen-2-yl)methylene)-1,3-di(thiophen-2-yl)-4*H*-cyclopenta[*c*]thiophene-4,6(5*H*)-dione (**6**)

A mixture of 5 (1.91 g, 3.0 mmol), Pd(PPh_3_)Cl_2_ (70 mg, 0.10 mmol), and tributyl(2-thienyl)stannane (2.46 g, 6.6 mmol) in dry THF (25 mL) was heated to reflux for 24 h. After cooling down to room temperature under an argon atmosphere, the reaction mixture was concentrated under reduced pressure and then purified by silica gel column chromatography to afford compound **6** as a solid (1.98 g, 81.5%). ^1^ H NMR (400 MHz, CDCl_3_, ppm): δ 8.16–8.12 (m, 2H), 7.89 (s,1H), 7.70 (s, 1H), 7.42–7.41 (m, 2H), 7.15–7.11 (m, 2H), 2.83 (t, 2H), 2.54 (t, 2H), 1.72 (m, 2H), 1.59 (m, 2H), 1.34 (m, 12H) ,0.99 (m, 6H). ^13^C NMR (100 MHz, CDCl_3_, ppm): δ 183.21, 144.67, 141.11, 138.71, 137.97, 137.95, 136.15, 136.07, 133.67, 133.40, 130.86, 129.96, 128.43, 31.75, 31.64, 31.58 , 30.58, 30.31, 29.27, 29.19, 27.88, 26.35, 22.70, 22.64, 14.15. HRMS (*m*/*z*, EI+) calcd for C_32_H_34_O_2_S_4_ 578.145, found 578.142.

#### 3.2.7. Synthesis of 1,3-Bis(5-bromothiophen-2-yl)-5-((4,5-dihexylthiophen-2-yl)methylene)-4*H*-cyclopenta[*c*]thiophene-4,6(5*H*)-dione (**M1**) 

A portion of NBS (153 mg, 0.84 mmol) was added to a solution of 6 (500 mg, 0.86 mmol) in THF (10 mL). The reaction mixture was stirred at room temperature overnight in darkness. The reaction mixture was then poured into water (30 mL), extracted with diethyl ether (30 mL × 3), and the organic layer was washed with saturated brine, dried over anhydrous MgSO_4_, filtered, and concentrated under reduced pressure. Purification by column chromatography provided compound **M1** as a solid (503 mg, 79.4%). ^1^ H NMR (400 MHz, CDCl_3_, ppm): δ 7.81 (s,1H), 7.73–7.70 (m, 2H), 7.67 (s,1H), 7.04–7.01 (m, 2H), 2.83 (t, 2H), 2.54 (t, 2H), 1.73 (m, 2H), 1.61 (m, 2H), 1.33 (m, 12H), 0.89(m, 6H). ^13^C NMR (100 MHz, CDCl_3_, ppm): δ 180.90, 182.23, 177.43, 163.38, 157.96, 144.14, 141.29, 138.73, 138.28, 134.89, 134.77, 134.64, 134.60, 133.67, 131.06, 130.28, 129.69, 129.61, 116.38, 116.26, 34.16, 31.75, 31.64, 31.56, 30.56, 29.30, 29.22, 29.14, 27.89, 22.71, 22.66, 14.16, 14.07. HRMS (*m*/*z*, EI+) calcd for C_32_H_32_Br_2_O_2_S_4_ 735.962, found 735.963.

#### 3.2.8. Synthesis of 5-((4,5-Dihexylthiophen-2-yl)methylene)-1,3-bis(4-hexylthiophen-2-yl)-4*H*-cyclopenta[*c*]thiophene-4,6(5*H*)-dione (**7**)

To a solution of **5** (700 mg, 1.26 mmol) in dry THF (25 mL), was added Pd(PPh_3_)Cl_2_ (70 mg, 0.10 mmol) and tributyl(3-hexyl-2-thienyl)stannane (1.88 g, 5.05 mmol) under argon atmosphere. The reaction mixture was heated to reflux for 24 h. After cooling down to room temperature, the reaction mixture was concentrated under reduced pressure and then purified by silica gel column chromatography to afford compound **7** as a red-orange oil (780 mg, 82.9%). ^1^H NMR (300 MHz, CDCl_3_, ppm): δ 8.07 (s,1H), 7.98 (s, 2H), 7.80 (s,1H), 7.12 (s,2H), 2.93 (t, 2H), 2.75 (m, 4H), 2.64 (t, 2H), 1.77 (m, 8H), 1.42 (m, 24H), 0.99 (m, 12H). ^13^C NMR (75 MHz, CDCl_3_, ppm): δ 183.03, 182.36, 156.77, 144.78, 144.63, 144.25, 140.89, 138.95, 138.26, 137.41, 136.30, 136.10, 133.63, 133.20, 133.02, 130.99, 130.89, 130.76, 123.39, 123.28, 77.51, 77.09, 76.66, 31.73, 31.60, 31.51, 30.52, 30.44, 30.39, 29.26, 29.19, 29.10, 27.83, 22.69, 22.63, 14.19, 14.15, 13.67. HRMS (*m*/*z*, EI+) calcd for C_44_H_58_O_2_S_4_ 746.330, found 747.339.

#### 3.2.9. Synthesis of 1,3-Bis(5-bromo-4-hexylthiophen-2-yl)-5-((4,5-dihexylthiophen-2-yl)methylene)-4*H*-cyclopenta[*c*]thiophene-4,6(5*H*)-dione (**M2**)

A portion of NBS (153 mg, 0.84 mmol) was added to a solution of **7** (300 mg, 0.40 mmol) in THF (10 mL). The reaction mixture was stirred at room temperature overnight in darkness. The reaction mixture was then poured into water (30 mL), extracted with diethyl ether (30 mL × 3), and the organic layer was washed with saturated brine, dried over anhydrous MgSO_4_, filtered, and concentrated under reduced pressure. Purification by column chromatography provided compound **M2** as a red oil (280 mg, 77.7%). ^1^H NMR (300 MHz, CDCl_3_, ppm): δ 7.83 (s,1H), 7.71 (m, 2H), 7.56 (s,1H), 2.84 (t, 2H), 2.57 (m, 6H), 1.74 (m, 2H), 1.60 (m, 6H), 1.33 (m, 24H), 0.92 (m, 12H). ^13^C NMR (75 MHz, CDCl_3_, ppm): δ 182.92, 182.22, 157.53, 144.62, 143.55, 143.35, 141.17, 139.17, 138.44, 137.92, 134.97, 134.83, 133.64, 132.96, 132.80, 130.46, 130.09, 129.74, 113.49, 113.25, 77.35, 77.03, 76.71, 31.68, 31.62, 31.57, 31.47, 30.50, 29.65, 29.55, 29.51, 29.23, 29.19, 29.14, 28.97, 28.93, 27.84, 22.63, 22.59, 14.11, 14.08. HRMS (*m*/*z*, EI+) calcd for C_44_H_56_Br_2_O_2_S_4_ 902.153, found 903.161.

#### 3.2.10. Synthesis of 5-((4,5-Dihexylthiophen-2-yl)methylene)-1,3-bis(6-octylthieno [3,2-b]thiophen-2-yl)-4*H*-cyclopenta[*c*]thiophene-4,6(5*H*)-dione (**8**)

A mixture of 5 (0.40 g, 0.70 mmol), Pd(PPh_3_)Cl_2_ (50 mg, 0.06 mmol), and trimethyl(6-octylthieno [3,2-*b*]thien-2-yl)stannane (1.20 g, 2.81 mmol) in dry THF (20 mL) under argon atmosphere was heated to reflux for 24 h. After cooling down to room temperature, the reaction mixture was concentrated under reduced pressure and then purified by silica gel column chromatography to afford compound **8** as a red-orange oil (380 mg, 59.4%). ^1^H NMR (400 MHz, CDCl_3_, ppm): δ 8.63 (s,1H), 8.48 (s, 1H), 7.74 (s,1H), 7.56 (s, 1H), 7.02 (s, 2H), 2.82 (t, 2H), 2.64 (m, 4H), 2.51 (t, 2H), 1.74 (m, 8H), 1.35 (m, 32H), 0.90 (m, 12H). ^13^C NMR (100 MHz, CDCl_3_, ppm): δ 182.84, 182.22, 156.88, 144.35, 141.49, 141.24, 140.81, 139.84, 139.66, 138.93, 137.38, 136.49, 136.37, 135.04, 135.00, 134.46, 134.36, 133.63, 130.74, 124.43, 124.34, 123.18, 122.78, 77.35, 77.03, 76.72, 31.91, 31.70, 31.59, 31.49, 30.49, 29.83, 29.72, 29.47, 29.41, 29.31, 29.21, 29.18, 28.59, 27.85, 22.71, 22.66, 22.63, 14.15, 14.11. MS (MALDI-TOF): [M]+ *m*/*z* calcd for C_52_H_66_O_2_S_6_ 914.339, found 915.394.

#### 3.2.11. Synthesis of 1,3-Bis(5-bromo-6-octylthieno [3,2-*b*]thiophen-2-yl)-5-((4,5-dihexylthiophen-2-yl)methylene)-4*H*-cyclopenta[*c*]thiophene-4,6(5*H*)-dione (**M3**)

A portion of NBS (188 mg, 1.04 mmol) was added to a solution of **8** (380 mg, 0.42 mmol) in THF (10 mL). The reaction mixture was stirred at room temperature overnight in darkness. The reaction mixture was then poured into water (30 mL), extracted with diethyl ether (30 mL × 3), and the organic layer was washed with saturated brine, dried over anhydrous MgSO_4_, filtered, and concentrated under reduced pressure. Purification by column chromatography provided compound **M3** as a red oil (240 mg, 62.2%). ^1^H NMR (400 MHz, CDCl_3_, ppm): δ 8.53 (s,1H), 8.36 (s, 1H), 7.59 (s,1H), 7.44 (s, 1H), 2.81 (t, 2H), 2.58 (m, 4H), 2.51 (t, 2H), 1.65 (m, 8H), 1.33 (m, 32H), 0.90 (m, 12H). ^13^C NMR (100 MHz, CDCl_3_, ppm): δ182.43, 181.77, 157.05, 144.16, 140.79, 140.20, 139.77, 139.50, 139.03, 138.15, 137.96, 137.06, 135.65, 135.48, 134.09, 133.83, 133.53, 130.19, 122.58, 122.18, 113.16, 113.09, 77.35, 77.03, 76.71, 34.68, 31.92, 31.72, 31.62, 31.47, 30.41, 29.72, 29.51, 29.41, 29.33, 29.25, 29.02, 27.97, 27.90, 25.29, 22.71, 22.66, 20.71, 14.16, 14.14. MS (MALDI-TOF): [M]+ *m*/*z* calcd for C_52_H_64_Br_2_O_2_S_6_ 1070.160, found 1070.262.

#### 3.2.12. Synthesis of IND-T-BDTF, IND-HT-BDTF and IND-OTT-BDTF

In a Schlenk flask, the monomers BDTF (0.25 mmol) and **M1**, **M2**, or **M3** (0.25 mmol), and Pd(PPh_3_)_4_ (5%) were dissolved in dry toluene (5 mL). The reaction mixture was stirred at 110 °C for 24 h under an argon atmosphere. Then, 2-tributylstannylthiophene (0.1 mL) and 2-bromothiophene (0.1 mL) were consecutively added as end-capping agents with an interval of 2 h. The polymer solution was cooled to room temperature and precipitated from methanol. The crude polymer was collected by filtration, and further purified by Soxhlet extraction. Finally, the polymer was collected from the chloroform fraction by precipitation from methanol and dried under a vacuum. IND-T-BDTF (210 mg, 60.5%); ^1^H NMR (400 MHz, CDCl_3_, ppm): δ 2.87 (s), 1.72–0.90 (m). GPC: Mn = 4319, PDI = 2.15. IND-HT-BDTF (220 mg, 63.5%); ^1^H NMR (400 MHz, CDCl_3_, ppm): δ 2.87 (s), 1.72–0.90 (m). GPC: Mn = 7494, PDI = 1.62. IND-OTT-BDTF (225 mg, 58.1%); ^1^H NMR (400 MHz, CDCl_3_, ppm): δ 3.03 (s), 1.54–0.96 (m). GPC: Mn = 22531, PDI = 2.42.

### 3.3. Fabrication and Analysis of PSCs

Inverted-type PSCs devices with a structure of ITO/ZnO (25 nm)/Polymer:Y6BO (80 nm)/MoO_3_ (3 nm)/Ag (100 nm) were fabricated. The ZnO layer was deposited on the ITO by sol-gel process giving a film of 25-nm-thick. The ZnO solution was prepared by dissolving 0.164 g of zinc acetate dehydrate and 0.05 mL of 2-aminoethanol in 1mL of methoxyethanol and then the corresponding solution was stirred for 30 min before the film deposition. The active layer was fabricated by spin-coating using a solution of the polymeric donor and Y6BO acceptor in chloroform with 0.5% of 1-chloronaphthalene (CN) as a processing additive. The blended solution was previously filtered using a 0.5 µm PTFE membrane filter. The active layer was heat treated at 110 °C for 10 min in the glove box. Then, a MoO_3_ layer (3-nm-thick) and an Ag layer (100-nm-thick), were consecutively deposited using a shadow mask with a device area of 0.09 cm^2^ by thermal evaporation at 2 × 10^−6^ Torr. The photovoltaic characteristics of the corresponding devices were evaluated using a KEITHLEY Model 2400 source-measure unit under AM 1.5G illumination at 100 mW/cm^2^ from a 150 W Xenon lamp. 

### 3.4. Fabrication of Hole- and Electron-Only Devices

Hole-only devices with the structure: [ITO/polymer:Y6BO/Au (50 nm)] and electron-only devices with the structure: [ITO/ZnO (25 nm)/polymer:Y6BO/Al (50 nm)], have been fabricated to investigate the hole and electron mobility.

## 4. Conclusions

In conclusion, the rational design of 4*H*-cyclopenta[*c*]thiophene-4,6(5*H*)-dione (IND) derivatives is to develop novel polymer donors. We created a new class of polymer donors called IND-T-BDTF, IND-HT-BDTF, and IND-OTT-BDTF, in which BDTF as donating unit is connected to the IND derivative electron-accepting unit via thiophene (T) or thieno [3,2-*b*]thiopene (TT) bridges. The energy levels of the polymers did not show discernible change upon replacement in the π-bridge. The absorption range of the polymer donors based on IND in this study showed 400~800 nm, which complimenting the absorption of Y6BO (600~1000 nm). The introduction of the hexyl group on the thiophene bridge improves the absorption coefficient in the ICT region. Further absorption coefficient improvement in the ICT region can be achieved upon replacement in the π-bridge from thiophene to thieno [3,2-*b*]thiopene (TT). The excitons were almost quenched in the blend films with Y6BO, indicating the charge transfer and exciton dissociation in the blend films have effectively occurred. In addition, the hole/electron mobility and ratios improved upon changes in the π-bridge. In conclusion, the PCEs of the PSCs were in the order of IND-T-BDTF (3.18%) < IND-HT-BDTF (7.23%) < IND-OTT-BDTF (11.69%). The highest PCE was obtained in the PSCs based on IND-OTT-BDTF.

## Data Availability

The data is available on request from the corresponding author.

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
