# Peer review of "Medium Bandgap Polymers for Efficient Non-Fullerene Polymer Solar Cells—An In-Depth Study of Structural Diversity of Polymer Structure"

_ijms, 2022, doi:10.3390/ijms24010522_

Round 1

Reviewer 1 Report

The manuscript entitled: “Medium bandgap polymers for efficient non-fullerene polymer solar cells; An in-depth study of structural diversity of polymer structure” submitted to International Journal of Molecular Sciences as Article describes design of three 4H-cyclopenta[c]thiophene-4,6(5H)-dione derivatives as donor polymers for inverted polymeric solar cells. Authors provide vast amount of theoretical and experimental data, including: multistep synthesis of polymeric compounds, 1h and 13C NMR, TGA of pure compounds; electro-optical studies of thin layers. Finally, the results for constructed solar cells are presented.

The topic presented in the manuscript discuss important aspects of polymeric solar cells based on non-fulleren acceptor and new polymeric donor. In my opinion described results could be of interest for scientific community. However, due to some mayor issues, I do not recommend this article for publishing in its current form.  

The list of issues that requires authors attention are present in form of comments in attached PDF file.

Author Response

Responses to the reviewer's comments are attached separately.

Author Response

(The authors gave the same response as above.)

Reviewer 3 Report

The research article titled "Medium bandgap polymers for efficient non-fullerene polymer 2 solar cells; An in-depth study of structural diversity of polymer 3 structure" is very interesting and it presents fundamental information on new polymer donor. 

Abstract: Abstract is well written  and conclusive 

Introduction: This section is brief , yet complete in all aspects. The authors cited new references mostly from the last 5 years. 

Results and discussion: The sections are presented in comprehensive manner. 

Conclusion : Conclusion is supported by the results presented in the draft

Overall the manuscript brings novel idea and it is worth for publication . 

Author Response

(The authors gave the same response as above.)

Round 2

Reviewer 1 Report

The manuscript entitled: “Medium bandgap polymers for efficient non-fullerene polymer solar cells; An in-depth study of structural diversity of polymer structure” submitted to International Journal of Molecular Sciences as Article describes design of three 4H-cyclopenta[c]thiophene-4,6(5H)-dione derivatives as donor polymers for inverted polymeric solar cells. Authors provide vast amount of theoretical and experimental data, including: multistep synthesis of polymeric compounds, 1h and 13C NMR, TGA of pure compounds; electro-optical studies of thin layers. Finally, the results for constructed solar cells are presented.
The article includes responses to reviewers comments form the first round of review process.

After careful analysis of changes done by authors in this current version of the article I do recommend this article for publishing in its current form.